# A Rapid Prototyping Approach for Multi-Material, Reversibly Sealed Microfluidics

**DOI:** 10.3390/mi14122213

**Published:** 2023-12-07

**Authors:** Michael Halwes, Melanie Stamp, David J. Collins

**Affiliations:** 1Department of Biomedical Engineering, University of Melbourne, Melbourne 3010, Australia; mhalwes@student.unimelb.edu.au (M.H.); melanie.stamp@unimelb.edu.au (M.S.); 2Graeme Clark Institute for Biomedical Engineering, University of Melbourne, Melbourne 3010, Australia

**Keywords:** microfluidics, rapid prototyping, additive manufacturing, lab-on-a-chip, organ-on-chip

## Abstract

Microfluidic organ-on-chip models recapitulate increasingly complex physiological phenomena to study tissue development and disease mechanisms, where there is a growing interest in retrieving delicate biological structures from these devices for downstream analysis. Standard bonding techniques, however, often utilize irreversible sealing, making sample retrieval unfeasible or necessitating destructive methods for disassembly. To address this, several commercial devices employ reversible sealing techniques, though integrating these techniques into early-stage prototyping workflows is often ignored because of the variation and complexity of microfluidic designs. Here, we demonstrate the concerted use of rapid prototyping techniques, including 3D printing and laser cutting, to produce multi-material microfluidic devices that can be reversibly sealed. This is enhanced via the incorporation of acrylic components directly into polydimethylsiloxane channel layers to enhance stability, sealing, and handling. These acrylic components act as a rigid surface separating the multiple mechanical seals created between the bottom substrate, the microfluidic features in the device, and the fluidic interconnect to external tubing, allowing for greater design flexibility. We demonstrate that these devices can be produced reproducibly outside of a cleanroom environment and that they can withstand ~1 bar pressures that are appropriate for a wide range of biological applications. By presenting an accessible and low-cost method, we hope to enable microfluidic prototyping for a broad range of biomedical research applications.

## 1. Introduction

For years, microfluidic organ-on-chip devices have advanced investigations into tissue development, disease mechanisms, and drug discovery [1]. These devices typically consist of a soft-polymer layer containing the microfluidic features which is then irreversibly bonded to a hard substrate before being connected to pumps for use. As the number of applications for organ-on-chip systems continues to increase, reversible sealing approaches are garnering increasing attention [2,3,4]. Reversible bonds between microfluidic layers and the underlying substrate can be created via vacuum pressure [5], magnetism [6], or direct mechanical clamping [7,8]. Commercially available systems [9] and research prototypes [10] have demonstrated the benefits of so-called ‘Lock-and-Play’ devices. Cell-laden scaffolds can be directly deposited onto, for example, a coverslip or membrane before being sealed within a microfluidic compartment for perfusion and subsequent culture, reducing the risks associated with loading cell suspensions into the device [11]. Additionally, retrieving cells grown within the devices for downstream analysis reduces the risk that fragile samples become damaged only after extended (and expensive) culturing processes [12].

Another key benefit of reversible sealing techniques is the liberty they grant designers to explore and integrate materials besides polydimethylsiloxane (PDMS), an elastomer often used for creating microfluidic devices. The difficulties in scaling up PDMS manufacturing processes and the ability of the PDMS polymer matrix to absorb small molecules have confounded efforts to employ microfluidics in high-throughput screening for pre-clinical drug development [13,14,15]. Moreover, plasma treatment of PDMS damages the polymer surface such that an irreversible bond can only be created once, and disassembling the device often leaves residual PDMS on the substrate [16]. However, PDMS continues to have significant value in the context of academic research or early-stage prototyping. PDMS is transparent, biocompatible, easy to bond to glass, quick to fabricate, and can be used as a mold to replicate features at the level of nanometers [17]. Specifically in the context of reversible sealing techniques, the elastomeric nature of PDMS makes it an ideal candidate for creating mechanical seals. Many examples of microfluidic devices presented in the literature fail to exploit this characteristic when designing the fluidic interconnects between the device and external pumps. Instead, alternative methods are often used in which a second batch of PDMS is cast around the original microfluidic device [18]. These techniques prolong the time before the device can be used and make the device more cumbersome to handle, but they are often preferred because of the lack of reliable and accessible commercial solutions. Ultimately, the compelling benefits of PDMS means it will likely continue to be used for microfluidic device fabrication, even as researchers exploit rapid prototyping techniques, for example, 3D printing and laser cutting/engraving, to explore novel designs.

Three-dimensional printing has been used to directly fabricate microfluidic devices for over a decade [18]. While the direct printing of PDMS devices is also feasible [19], more attention has been invested in indirect microfluidic fabrication, where master molds are 3D printed and used to cast PDMS devices [20]. Compared to traditional microfluidic soft lithography, 3D-printing-based approaches allow researchers to more quickly and cost-effectively iterate through microfluidic designs while avoiding the need for a cleanroom environment [21]. A wide variety of 3D printing techniques have been studied in the context of microfluidic fabrication, but for applications requiring fine detail in the sub-200 μm range, light-based techniques such as stereolithography (SLA) or digital light processing (DLP) have tended to perform better than extrusion-based processes like fused deposition modelling (FDM) [22]. Additionally, the rising number and decreasing cost of consumer-grade SLA and DLP printers mean that these techniques continue to become more accessible to a wider range of research groups.

Like 3D-printing, laser cutting has enabled microfluidic research on a much broader range of materials than PDMS [23,24,25,26]. By cutting or etching channels in thin films, multiple layers can be stacked to create complex systems [27]. Alternatively, thicker pieces of acrylic can be cut and etched to create both channels and reservoirs at once [28]. However, depending on the material and thickness used, recreating features smaller than 100–150 μm can be difficult [29,30]. Separately, laser cutting and 3D printing are proven techniques in microfluidic fabrication [31]. In tandem, the two techniques can complement each other to make research more accessible and enable faster design iterations. Specifically, by combining PDMS microfluidic features with laser-cut components, we can exploit the elastomeric properties of PDMS while also retaining the advantages that rigid components offer in designing mechanical seals. On the other hand, 3D-printed parts can enable benchtop fabrication of these multi-material microfluidic devices and make connecting to standard microfluidic fittings more straightforward.

Here, we present a novel approach by which laser cutting and 3D printing can be combined to create multi-material microfluidic devices for reversible sealing applications. Uniquely, laser-cut acrylic was embedded within PDMS while simultaneously casting features from 3D-printed molds. Once the devices were fabricated, a 3D-printed clamping setup was used to conduct burst pressure measurements and establish the feasibility of the approach. Critically, by embedding a rigid acrylic component within the PDMS matrix of the device, the clamping force could be equally distributed over multiple microfluidic subunits. Additionally, the embedded acrylic provided a clamping surface for sealing against customized well reservoirs or pump adapters independent of the seal created between the microfluidic features and the glass substrate. We demonstrate that reliable devices can be fabricated quickly and reproducibly with the aim of reducing barriers to entry for microfluidic design and fabrication. We then highlight how the novel aspects of the devices enable greater design flexibility, showcasing the value that rapid prototyping techniques can offer in microfluidics research.

## 2. Materials and Methods

### 2.1. Approach and System Overviews

To demonstrate the approach, a microfluidic device and two sets of assemblies were designed and fabricated, comprising casting/molding (Figure 1A) and clamping (Figure 1B) systems. The microfluidic device was designed as an array of three PDMS ‘chips’ (Sylgard 184) encasing an acrylic backbone that added rigidity to the device and more equally distributed the clamping force. Either laser-cut fluid wells or a 3D-printed pump adapter could be assembled and fixed to the top of the device for passive or active flow applications, respectively.

The molding assembly for creating the PDMS–acrylic devices consists of the base mold containing the negative of the microfluidic features, a mold jig for controlling the relative position of the mold and acrylic backbones, acrylic spacers for controlling the height of PDMS above the acrylic backbone, an acrylic template for controlling the planarity of the top PDMS surface, and a holder that clamps the jig and mold together. Once the devices are cast, they are placed in a clamping assembly made up of a housing with posts for two carriages that are secured with set screws. A cam is mounted onto each carriage to provide the clamping force to the microfluidic device, securing it to a microscope slide for later testing.

### 2.2. Part Fabrication

A commercial resin-based 3D printer (Formlabs Form3, Somerville, MA, USA) and laser cutter (Trotec Speedy 100, Marchtrenk, Austria) were used for part fabrication. All 3D-printed parts were post-processed according to the manufacturer’s instructions prior to being baked at 80 °C for 48 h to remove all residues which would have interfered with the PDMS curing process [32]. All laser-cut components were wiped down with ethanol prior to use.

For casting the microfluidic devices, Sylgard 184 was mixed at a 10:1 ratio, poured into the assembled mold, and cured at 80 °C overnight. The resulting microfluidic devices were disassembled from the mold, cut out, and made ready for use; the inlet and outlet pathways were included in the 3D-printed mold to remove the need for hole punching. For more information, please refer to the Appendix A.

### 2.3. Surface Roughness Measurements

To assess the quality of the post-processed 3D-printed molds and therefore the resulting microfluidic channels, surface roughness measurements were taken using an optical profilometer (Bruker ContourGT, Billerica, MA, USA). A window of 640 × 480 px (corresponding to an area of 317 × 268 μm) was measured at three representative points on three separate molds, giving a total of 9 measurements. The two-dimensional roughness S_a_ was then measured in Gwyddion v2.63 (Czech Metrology Institute, Jihlava, Czechia) following mean plane subtraction and averaged for all measurements.

### 2.4. Channel Height Measurements

To measure the height of the molded microfluidic channels under clamped and unclamped conditions, the microfluidic devices were assembled into the clamping setup and placed on an inverted microscope (Zeiss Observer.Z1, Oberkochen, Germany). The height of the channels in the left, center, and right chips was then measured by calculating the difference between the objective’s Z-position when it was focused on the glass slide and the tops of the microfluidic features. Once the unclamped height was measured, the cams were turned 40° relative to vertical, corresponding to a clamping distance of 0.424 mm, and the clamped height was measured through the viewports in the bottom of the clamping assembly holder. The channel heights were measured both when the acrylic fluid wells and the 3D-printed pump adapter were placed above the microfluidic devices. Measurements were taken from five separate microfluidic devices.

### 2.5. Burst Pressure Measurements

Burst pressure measurements were recorded to validate the clamp-based microfluidic performance. The test circuit consisted of a syringe pump (KD Scientific, Holliston, MA, USA) which slowly injected water mixed with green food dye into the sealed chips, while the pressure was measured using an inline pressure sensor (Fluigent Pressure Unit XL, Paris, France). The pressure at which a leak formed was recorded and compared between the left, center, and right chips of the devices. Five devices were tested, each containing three chips, with the burst pressure of each chip measured three times (i.e., three technical replicates) before being averaged.

## 3. Results and Discussion

### 3.1. Surface Roughness

The surface profile of the 3D-printed molds can be seen in Figure 2. The mean two-dimensional surface roughness of the molds, S_A_, was 6.656 μm (σ = 2.385 μm). The surface roughness of the mold was measured as an indirect measure of the real surface roughness of the cast PDMS microfluidics, due to the improved imaging quality of opaque samples in optical profilometry. As Sylgard 184 is able to recreate details with feature sizes <100 nm [32], we feel it is appropriate to assume that the PDMS surface would follow an equivalent surface profile on the microscale. A recent study investigating PDMS microfluidics cast from 3D-printed molds showed that the surface roughness of cast PDMS was on the same order of magnitude as (and indeed somewhat lower than) the surface roughness of the master molds [33]. Therefore, we regarded the surface roughness of the mold as a worst-case scenario measurement for the PDMS surface roughness. As the focus of this work was to demonstrate the principle of using multiple rapid prototyping techniques in combination to produce microfluidic devices, additional surface treatments were not explored. However, multiple post-processing treatments can decrease surface roughness [34], such as sanding, chemical mechanical polishing [35], or coating with materials such as nail polish [33], parylene-C [36], or omniphobic lubricants [37].

Generally, stereolithography printing is becoming more widely available and accepted among microfluidics researchers. However, leaching of compounds like residual monomers and photoinitiators from 3D-printed parts into cell culture has been shown to negatively impact cell viability and proliferation, reducing the biocompatibility of devices depending on their intended applications [38,39]. Our approach avoids this risk; in the clamp-based setup using the laser-cut acrylic wells, media would only contact glass, PDMS, or acrylic aside from the cell culture itself. However, leaching of resin-based compounds can also occur indirectly through the PDMS matrix cast on 3D-printed molds [40]. A recent investigation into leachates using a gas- and liquid-chromatography-based approach coupled with mass spectroscopy provided two important findings. Firstly, chemicals known to cause cytotoxicity under certain circumstances leached into PDMS samples cast from both 3D-printed and traditional SU-8 molds. Secondly, HeLa cells could successfully be cultured in the resulting PDMS devices in vitro. While the effects of 3D-printed leachates must be carefully evaluated, particularly for high-abundance leachates such as polypropylene glycol, several post-processing methods have been shown to reduce leachate concentration to safer levels, including incubating in water [41], treating with ethanol [42], coating with parylene-c [43], or autoclaving [44].

### 3.2. Channel Height

The channel elements (device in Figure 3A) are fabricated with a uniform height, and an acrylic laser-cut sheet is integrated into the structure of the PDMS element in order to enhance stability across the clamped channels. The compression forces are thus translated through the clamping mechanism via this acrylic layer into the PDMS channels, resulting in deviations from the design height in the realized fluidic volumes. The bar graph in Figure 3C depicts the channel heights under the conditions in which the microfluidic channels were either unclamped, clamped under acrylic wells, or clamped under the 3D-printed adapter for standard microfluidic fittings. Whereas the channel heights for the unclamped setting fall in close line with one another (range: 14.42 μm, σ = 7.28 μm), clear irregularities can be seen in both clamped conditions. The higher variability of the channels when clamped beneath the acrylic wells (range: 71.50 μm) compared to the adapter (range: 44.50 μm) can most likely be attributed to the increased rigidity of the 3D-printed resin compared to the 6 mm thick laser-cut acrylic sheet. Increased deflection of the wells on the ends (compared to the adapter) would contribute to increased variation in channel height.

Among all three conditions, a linear trend in variability can be seen when comparing the left, center, and right chip positions. The effect is exaggerated in the clamped condition (σ_L_ = 57 μm and σ_R_ = 34 μm for the wells) compared to the unclamped condition (σ_L_ = 15 μm and σ_R_ = 7 μm) and can be attributed to warpage of the mold on which the PDMS microfluidics were cast. To avoid the possibility of collapsed microfluidic channels, a consistent 40° cam angle was chosen as the clamping condition for burst pressure measurements as well. Ultimately, the non-significant difference (*p* = 0.30) between the left, center, and right mean channel heights in the unclamped condition shows that the molding process presented here can be used to create reliable microfluidic features.

### 3.3. Burst Pressure

The burst pressure measurements, sorted by chip position on the microfluidic device, are shown in Figure 4. There was a high range of achievable burst pressures, due possibly to tolerances in the clamping distance in the clamping assembly or lateral positioning of the microfluidic devices in the assembly. However, the clamping approach nevertheless demonstrated mean burst pressures exceeding 1.2 bar across five devices. Additionally, the minimum burst pressure across all trials each exceeded 500 mbar, which would be more than sufficient to replicate physiological pressures investigating, for example, respiratory or vascular phenomena [45,46].

When comparing results between separate devices, warpage of the mold most likely contributed to variability in device performance. Assuming that the height of the cam above the device, and hence the clamping distance, is well controlled, the resulting curvature of the PDMS–acrylic device, both unclamped and assembled, depends on the warpage of both the acrylic backbone and the 3D-printed molds. The laser power, pulse rate, and speed were controlled on the laser cutter such that warpage of the acrylic sheet due to thermal stresses was avoided. However, warpage of the 3D-printed mold due to shrinkage strains created during the standard and high temperature (80 °C) post-processing steps consistently led to non-planarity of the mold surface [47,48]. Controlling for warp in complex designs a priori is a non-trivial issue [49] and is instead often accomplished through design iterations. In the future, alternative strategies for controlling thermal warpage, e.g., controlling the rate at which the mold is heated and cooled, could be explored as an improvement of microfluidic device quality.

Different methods can be used to bond multiple microfluidic layers to one another or to seal the final device to a standard substrate. These include surface activation using air plasma or corona treatment, chemical gluing using silanizing agents, or adhesives to bond PDMS to glass, polymer, or other PDMS substrates [16]. While these techniques can often be used to achieve higher burst pressures than reported here, they often introduce additional costs in the form of reagents, equipment/facility infrastructure, and time. Moreover, the surface roughness of PDMS microfluidics cast from 3D-printed molds often reduces the final bond strength, necessitating an additional manufacturing step where the cavities on the surface are filled in with liquid PDMS [50]. Another drawback is that these methods ultimately consume the substrate as well as the microfluidic layer; residual PDMS on the substrate surface precludes repeated uses. By sealing the devices to the substrates reversibly, both could be reused, and a wider range of substrate materials could be used. This could enable easier integration of electrode sensors or sensitive materials, including multi-electrode arrays or organic electrochemical transistors, into microphysiological systems [51,52].

In the devices presented here, we have used laser-cut and 3D-printed components to enable novel microfluidic device designs. A comparison with another multi-material, reversibly sealed microfluidic device from the literature helps to highlight the benefits our approach offers in designing the mechanical seals of the device [53]. While the device presented by Pitingolo et al. also uses acrylic substrates to transmit a clamping force onto a layer of PDMS to seal their device, the PDMS was deposited onto the acrylic substrate by spin-coating, and the acrylic substrate defined a single fluidic connection to external tubing. By using 3D-printed components to embed the acrylic within the PDMS matrix, our devices allow for greater flexibility in the design of the fluidic connections between the microfluidic channels and external reservoirs. In the setup for the burst pressure measurements, the embedded acrylic provides a rigid surface for both the seal between the microfluidic channels and the glass slide and the seal between the device and the pump adapter. This allows both seals to be designed independently from each other, as the clamping force is first redistributed from the smaller sealing edge on the pump adapter (see Figure 3A) to the broader face of the acrylic backbone before compressing the microfluidic channels against the glass substrate. The benefit of this design independence was reflected in device performance during the burst pressure measurements; leaks always formed between the microfluidic channel and the glass substrate, whereas leaks never formed between the pump adapter and the top PDMS surface, even though a single clamping force was applied to both seals. In the future, the design of this acrylic backbone could be customized for different microfluidic geometries in line with standardized guidelines for designing microfluidic seals [54].

## 4. Conclusions

The method we present here serves as a proof of concept showcasing the possibilities of combining rapid prototyping techniques for microfluidic device design; to our knowledge, this is the first example of a multi-material, reversibly sealed microfluidic device incorporating a rigid element embedded within the polymer matrix. This rigid element enabled reliable mechanical performance while also simplifying the design of the fluidic seals within the device. Moreover, these devices were produced without cleanroom equipment or standard photolithography. As the microfluidic market matures, many commercial partners are beginning to offer contract development options for customized microfluidics and organ-on-chip models. However, for early-stage design exploration or niche biological applications, many users will continue to fabricate lower-cost prototypes without being ‘locked in’ to a standardized design with potentially onerous design constraints. To address this user need, we produced microfluidic devices outside of a cleanroom environment, employing no chemical agents other than 3D-printing resins, PDMS, and ethanol, and requiring only a 3D printer, vacuum chamber, laser cutter, and oven. By directly integrating the acrylic backbone into the PDMS, the clamping force can be evenly distributed over an array of devices. At the same time, reliable compressive seals can still be formed between the device and the wells or pump adapter. Leaks in the burst pressure tests always formed between the PDMS and substrate rather than between the PDMS and pump adapter. Having a rigid component integrated into the PDMS also provides the advantage of handling and positioning the device more easily. Overall, we demonstrate how 3D printing and laser cutting can be used to improve the reliability of microfluidic prototyping. Our hope is that the simplicity of the methods presented here provide a useful example and toolbox for researchers hoping to employ rapid prototyping techniques in their microfluidic designs.

By combining increasingly common rapid prototyping workflows in new ways, we have shown that reversibly sealed microfluidic devices can be fabricated reproducibly and quickly at relatively low cost. Laser-cut components can be integrated with PDMS microfluidics that are capable of sustaining fluid pressures relevant for a wide array of biological applications. This proof of concept was accomplished without the need for traditional photolithography equipment, cleanroom conditions, or chemical treatments. At the same time, the accessibility of the approach still allows for additional surface treatment processes in the future, if required. The methods presented here can easily be integrated into a wide range of laboratory settings and adapted for different materials or designs such that researchers can quickly fabricate, test, and iterate their microfluidic prototypes.

## Figures and Tables

**Figure 1 micromachines-14-02213-f001:**
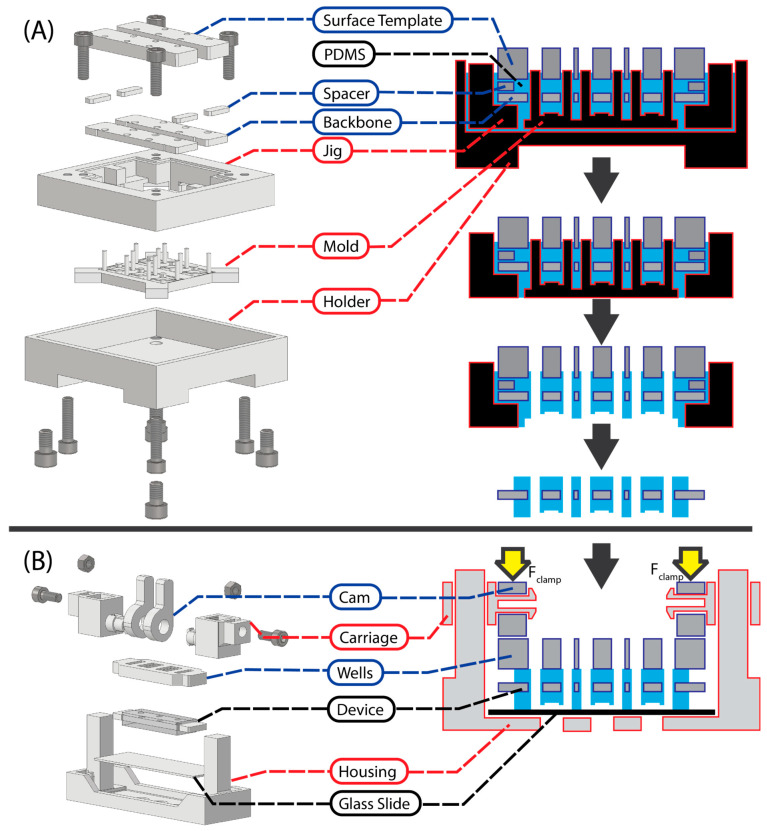
Exploded views of the rapid-prototyped assembly for manufacturing multi-material microfluidic devices. In the right-hand column, the stepwise process by which the microfluidic devices are manufactured and then assembled for testing is shown. (**A**) The casting assembly for embedding laser-cut acrylic backbones within the cast PDMS microfluidic device. (**B**) The clamping assembly used for sealing the microfluidic devices to a glass substrate, measuring the channel height, and performing burst pressure tests. For all component labels, a red outline indicates a 3D-printed component, a blue outline indicates a laser-cut component, and a black label indicates a consumable. For simplicity, the machine screws are not labelled.

**Figure 2 micromachines-14-02213-f002:**
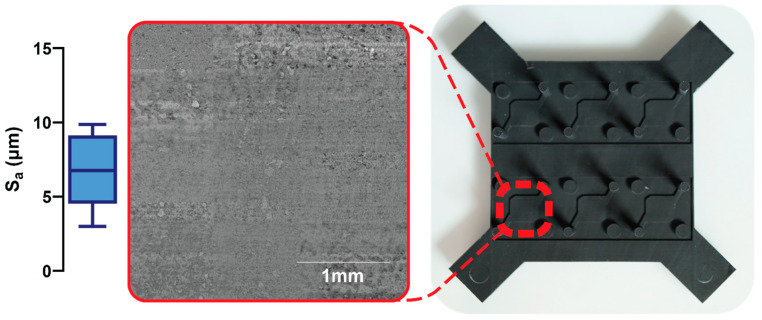
The 3D-printed mold for casting PDMS microfluidic features. The inset depicts a 3 × 3 mm FOV of the surface profile as obtained from optical profilometry. The box-and-whisker plot on the right shows the two-dimensional surface roughness, S_a_. Box limits depict 25th and 75th percentiles, horizontal line indicates median, and whiskers represent upper and lower limits.

**Figure 3 micromachines-14-02213-f003:**
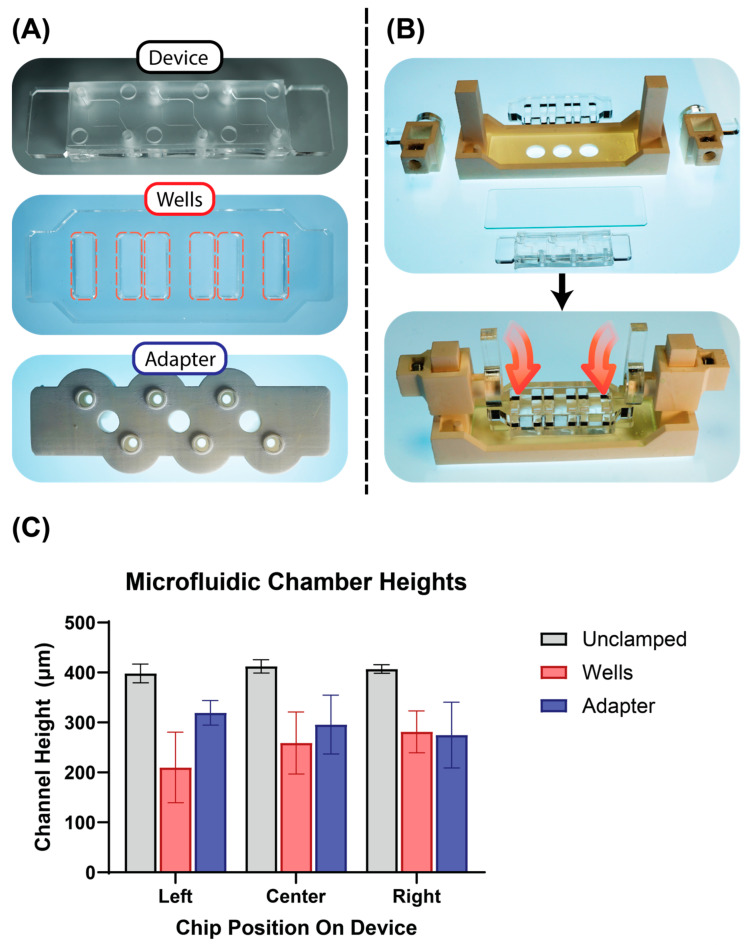
Overview of the system components and clamping measurement results. (**A**) From top to bottom: the microfluidic device as manufactured using the casting assembly, the laser-cut acrylic wells (outlined in red), and the 3D-printed pump adapter for standard microfluidic fittings (all images are bottom views of the respective components). (**B**) Overview of clamping assembly components with illustration of how components were assembled for chamber height measurements. (**C**) Bar graph showing the height of the microfluidic chamber under unclamped and clamped conditions. The color scheme of the bar corresponds to the color scheme of the outline of the images in (**A**).

**Figure 4 micromachines-14-02213-f004:**
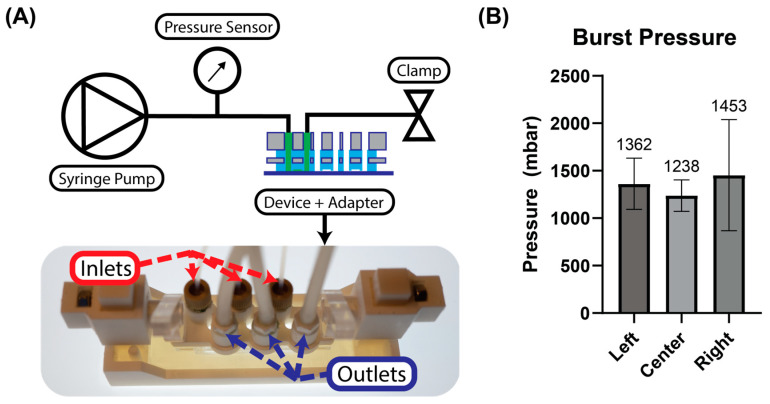
The microfluidic device setup for burst pressure measurements. (**A**) Top: fluid flow diagram of the burst pressure test setup. Bottom: The clamping setup with the pump adapter attached. Each inlet is fitted for a standard 1/4″-28 microfluidic nut-and-ferrule-style connector for 1/16″ OD PTFE tubing, and each outlet was fitted with a barbed connector for silicone tubing. After each chip was primed, the silicone tubing was clamped closely to conduct the burst pressure test. (**B**) The results of the burst pressure tests sorted by chip location (left, center, or right) on the device. The bars represent the mean value, with error bars showing SD. Each bar is labelled with the mean burst pressure.

## Data Availability

Data are contained within the article and Appendix A.

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
