# Peer review of "A Rapid Prototyping Approach for Multi-Material, Reversibly Sealed Microfluidics"

_micromachines, 2023, doi:10.3390/mi14122213_

Round 1
Reviewer 1 Report
Comments and Suggestions for Authors
This work is a very relevant example of chip development for organ-on-chip studies where reversible bonding is mandatory. This work is also an excellent example of the use of 3D printing and subsequent molding/overmolding for device development. The many references about potential drawbacks associated with 3D printing (toxicity, PDMS molding, etc...) asses for am extensive experience of the authors on the subject. In the introduction and in the conclusion, there is a very good discussion about the need for very low cost and affordable chip development tools for rapid prototyping without the need of costly cleanrooms, and similar equipments.
I recommend this paper for publication, with minor suggestions for corrections:
Line 50: replace invaluable by valuable? (the "in"vlauable reads very negative in my sense, but I am not a native English speaker)
Line 54: remove the second "as"
Line 102: Part fabrication: About the laser fabrication, I would add the main parameters, or refer to a paper that mentions them (same remark for lines 241-243)
Line 276-277: The authors mention "this is the 1st example of a reversibly sealed, multi-material microfluidics device produced without cleanroom equipment or standard photolithography". I think this is a bit presumptuous to say that. There have a quite a few PDMS devices loaded on CNC-milled molds and then clamped of CNC-machined sample holders. These past CNC-based solution were not as elegant as this one, but still.. they exist. Please rephrase (after 1 minute search I found at least 1 paper in this very topic: https://www.mdpi.com/2072-666X/13/10/1573)
Author Response
We thank both reviewers for their helpful and constructive feedback on our manuscript. We have integrated changes to the manuscript based on their comments and highlighted those changes in yellow. Additionally, we have included responses to the individual comments to explain these changes (original comments italicized). Overall, a common concern was the lack of clarity regarding the novelty of the approach we present. We have revised the abstract, introduction, and discussion sections of the manuscript to more clearly define the novel aspects of our approach and what value these aspects provide for microfluidics research. We again would like to thank the reviewers for their time and feedback, and we hope all questions or concerns have been addressed satisfactorily.
This work is a very relevant example of chip development for organ-on-chip studies where reversible bonding is mandatory. This work is also an excellent example of the use of 3D printing and subsequent molding/overmolding for device development. The many references about potential drawbacks associated with 3D printing (toxicity, PDMS molding, etc...) asses for am extensive experience of the authors on the subject. In the introduction and in the conclusion, there is a very good discussion about the need for very low cost and affordable chip development tools for rapid prototyping without the need of costly cleanrooms, and similar equipments.
I recommend this paper for publication, with minor suggestions for corrections:
Line 50: replace invaluable by valuable? (the "in"vlauable reads very negative in my sense, but I am not a native English speaker)
Response: We have revised the sentence to remove any ambiguity:
- “However, PDMS continues to have significant value in the context of academic research or early-stage prototyping.”
Line 54: remove the second "as"
Response: We apologize for the grammatical error, the line has been revised.
Line 102: Part fabrication: About the laser fabrication, I would add the main parameters, or refer to a paper that mentions them (same remark for lines 241-243)
Response: We have included a table detailing the laser cutting parameters in a supplementary information document, in line with the second reviewer’s comments.
Line 276-277: The authors mention "this is the 1st example of a reversibly sealed, multi-material microfluidics device produced without cleanroom equipment or standard photolithography". I think this is a bit presumptuous to say that. There have a quite a few PDMS devices loaded on CNC-milled molds and then clamped of CNC-machined sample holders. These past CNC-based solution were not as elegant as this one, but still.. they exist. Please rephrase (after 1 minute search I found at least 1 paper in this very topic: https://www.mdpi.com/2072-666X/13/10/1573)
Response: We appreciate the reviewer’s comment and apologize for the lack of clarity in our original submission. We have revised the quoted section as follows to highlight what we mean when discussing the novelty of the device we present. The reviewer is correct to cite CNC-based solutions as another example of multi-material devices. Our intention was to highlight the hybrid nature of the acrylic backbone integrated into the polymer matrix, rather than the environment in which the devices were made. We have revised the relevant lines to read as follows:
- “The method we present here serves as a proof of concept showcasing the possibilities of combining rapid prototyping techniques for microfluidic device design; to our knowledge, this is the first example of a reversibly sealed, multi-material microfluidic device incorporating a rigid element embedded within the polymer matrix. This rigid element enabled reliable mechanical performance while also simplifying the design of the fluidic seals within the device. Moreover, these devices were produced without cleanroom equipment or standard photolithography.”
Reviewer 2 Report
Comments and Suggestions for Authors
This manuscript reported a multi-material, reversibly, sealed microfluidics. This work made an improvement on the microfluidics and it was organized and written properly. However, the novelty should be clearly stated in the abstract and introduction. The comparison between current work and previous reports is suggested.
Major
1. It suggests adding more data details to clarify the novelty and enhanced performance.
2. Introduction. It should emphasize the novelty of the work and the challenges solved by the improved methods. Lines 42-55 should be shortened.
3. 2.2 Part Fabrication. It is a routine work and should be shorten or moved to supporting information.
4. Please add an application of the enhanced results.
Comments on the Quality of English LanguageEnglish should be polished by a native speaker.
Author Response
We thank both reviewers for their helpful and constructive feedback on our manuscript. We have integrated changes to the manuscript based on their comments and highlighted those changes in yellow. Additionally, we have included responses to the individual comments to explain these changes (original comments italicized). Overall, a common concern was the lack of clarity regarding the novelty of the approach we present. We have revised the abstract, introduction, and discussion sections of the manuscript to more clearly define the novel aspects of our approach and what value these aspects provide for microfluidics research. We again would like to thank the reviewers for their time and feedback, and we hope all questions or concerns have been addressed satisfactorily.
This manuscript reported a multi-material, reversibly, sealed microfluidics. This work made an improvement on the microfluidics and it was organized and written properly. However, the novelty should be clearly stated in the abstract and introduction. The comparison between current work and previous reports is suggested.
Major
- It suggests adding more data details to clarify the novelty and enhanced performance.
Response: We appreciate the reviewer’s comment and have made several adjustments to the manuscript and figures to attempt to clarify what we feel are the novel aspects of our approach and the advantages those aspects provide. We apologize for not emphasizing the utility of completely embedding the laser-cut component of the microfluidic device sufficiently in our original manuscript. As it regards the mechanical performance of the device, the rigid acrylic component helps to make the mechanical seals created between the layers of the device (substrate, microfluidic features, and pump adapter) more reliable. This is possible because the clamping force is effectively redistributed from the small, sealing edges of the pump adapter to the broader face of the acrylic backbone. The existence of this sealing edge was not well highlighted in the original submission, therefore we have changed the image of the pump adapter in Figure 3a to be a bottom view of the component. Additionally, we have added the following section to the Results and Discussion to explain the novelty and usefulness of our approach compared to previous literature.
- “In the devices presented here, we have used laser-cut and 3D printed components to enable novel microfluidic device designs. A comparison with another reversibly sealed, multi-material microfluidic device from the literature helps to highlight the benefits our approach offers in designing the mechanical seals of the device [54]. While the device presented by Pitingolo et al. also uses acrylic substrates to transmit a clamping force onto a layer of PDMS to seal their device, the PDMS was deposited onto the acrylic substrate by spin-coating, and the acrylic substrate defined a single fluidic connection to external tubing. By using 3D-printed components to embed the acrylic within the PDMS matrix, our devices allow for greater flexibility in the design of the fluidic connections between the microfluidic channels and external reservoirs. In the setup for the burst pressure measurements, the embedded acrylic provides a rigid surface for both the seal between the microfluidic channels and the glass slide and the seal between the device and the pump adapter. This allows both seals to be designed independently from each other, as the clamping force is first redistributed from the smaller sealing edged on the pump adapter (see Figure 3a) to the broader face of the acrylic backbone before compressing the microfluidic channels against the glass substrate. The benefit of this design independence was reflected in device performance during the burst pressure measurements; leaks always formed between the microfluidic channel and the glass substrate, whereas leaks never formed between the pump adapter and the top PDMS surface, even though a single clamping force was applied to both seals. In the future, the design of this acrylic backbone could be customized for different microfluidic geometries in line with standardized guidelines for designing microfluidic seals [55].“
- It should emphasize the novelty of the work and the challenges solved by the improved methods. Lines 42-55 should be shortened.
Response: We apologize for our lack of clarity in introducing the problems we aimed to solve with our approach, and have revised the abstract and introduction in an attempt to resolve this. Regarding lines 42-55, we have provided additional details from the literature that we hope clarify the relevance of PDMS and its properties to the current work:
- “Another key benefit of reversible sealing techniques is the liberty they grant designers to explore and integrate materials besides polydimethylsiloxane (PDMS), an elastomer often used for creating microfluidic devices. The difficulties in scaling up PDMS manufacturing processes and the ability of the PDMS polymer matrix to absorb small molecules have confounded efforts to employ microfluidics in high-throughput screening for pre-clinical drug development [13-15]. Moreover, plasma treatment of PDMS damages the polymer surface such that an irreversible bond can only be created once, and disassembling the device often leaves residual PDMS on the substrate [16]. However, PDMS continues to have significant value in the context of academic research or early-stage prototyping. PDMS is transparent, biocompatible, easy to bond to glass, quick to fabricate, and can be used as a mold to replicate features at the level of nanometers[17]. Specifically in the context of reversible sealing techniques, PDMS’ elastomeric nature makes it an ideal candidate for creating mechanical seals. Many examples of microfluidic devices presented in the literature fail to exploit this characteristic when designing the fluidic interconnects between the device and external pumps. Instead, alternative methods are often used in which a second batch of PDMS is cast around the original microfluidic device [18]. These techniques prolong the time before the device can be used and make the device more cumbersome to handle, but they are often preferred because of the lack of reliable and accessible commercial solutions. Ultimately, the compelling benefits of PDMS means it will likely continue to be used for microfluidic device fabrication, even as researchers exploit rapid prototyping techniques, for example 3D printing and laser cutting/engraving, to explore novel designs.”
Additionally, we have revised the last two paragraphs of the introduction to clarify the novelty of our approach and why this novelty can be helpful for designing and testing microfluidic devices:
- “Like 3D-printing, laser cutting has enabled microfluidic research in a much broader range of materials than PDMS[24-27]. By cutting or etching channels in thin films, multiple layers can be stacked to create complex systems[28]. Alternatively, thicker pieces of acrylic can be cut and etched to create both channels and reservoirs at once[29]. However, depending on the material and thickness used, recreating features smaller than 100-150μm can be difficult[30, 31]. Separately, laser cutting and 3D printing are proven techniques in microfluidic fabrication[32]. In tandem, the two techniques can complement each other to make research more accessible and enable faster design iterations. Specifically, by combining PDMS microfluidic features with laser-cut components, we can exploit the elastomeric properties of PDMS while also retaining the advantages that rigid components offer in designing mechanical seals. 3D-printed parts, on the other hand, can enable benchtop fabrication of these multi-material microfluidic devices and make connecting to standard microfluidic fittings more straightforward.
- Here, we present a novel approach by which laser cutting and 3D printing can be combined to create multi-material microfluidic devices for reversible sealing applications. Uniquely, laser-cut acrylic was embedded within PDMS while simultaneously casting features from 3D-printed molds. Once the devices were fabricated, a 3D-printed clamping setup was used to conduct burst pressure measurements and establish the feasibility of the approach. Critically, by embedding a rigid acrylic component within the PDMS matrix of the device, the clamping force could be equally distributed over multiple microfluidic subunits. Additionally, the embedded acrylic provided a clamping surface for sealing against customized well reservoirs or pump adapters independent of the seal created between the microfluidic features and the glass substrate. We demonstrate that reliable devices can be fabricated quickly and reproducibly with the aim of reducing barriers to entry for microfluidic design and fabrication. We go on to highlight how the novel aspects of the devices enable greater design flexibility, showcasing the value rapid prototyping techniques can offer in microfluidics research.”
- 2 Part Fabrication. It is a routine work and should be shorten or moved to supporting information.
Response: We sympathize with the reviewer’s desire for brevity, and have removed details from the Part Fabrication section regarding CAD design. In line with the first reviewer’s comments, we have also included a supplementary information document describing the laser cutter parameters used for part fabrication. However, concerning the high-temperature post-processing step, we feel it is important to highlight this in the main text of the work, as it represents an additional requirement for the 3D printed parts before they can be used with PDMS. In our experience, this need to heat treat the parts is often overlooked. Regarding the mixing ratio of Sylgard 184 and curing conditions for the microfluidic devices, we feel it is important to directly report these details in the main text, as the effects of mixing ratio and curing temperature on PDMS mechanical properties is often underappreciated between studies, possibly leading to complications when attempting to replicate findings. The revised Part fabrication section now reads as follows:
- “A commercial resin-based 3D printer (Formlabs Form3) and laser cutter (Trotec Speedy 100) were used for part fabrication. All 3D-printed parts were post-processed according to the manufacturer’s instructions prior to being baked at 80°C for 48 hours to remove all residues which would have interfered with the PDMS curing process[33]. All laser-cut components were wiped down with ethanol prior to use.
For casting the microfluidic devices, Sylgard 184 was mixed at a 10:1 ratio, poured into the assembled mold, and cured at 80°C overnight. The resulting microfluidic devices were disassembled from the mold, cut out, and ready for use; the inlet and outlet pathways were included in the 3D-printed mold to remove the need for hole punching.”
- Please add an application of the enhanced results.
Response: We appreciate the reviewer’s comment regarding additional applications to demonstrate the enhanced performance of our device. However, we feel that by providing additional clarity around the novelty of our approach and the design flexibility it offers microfluidics researchers, we have demonstrated the key advance our work offers. The purpose of this study was to demonstrate the proof of concept that combining 3D-printing and laser-cutting techniques with standard microfluidic materials can result in devices that are sufficiently reliable for biological applications without requiring more complex, costly, or time-consuming manufacturing/assembly equipment or reagents. Our intent is to first validate this with the characterization and burst pressure measurements made here, and that future work can go on to provide additional experiments in specific biological niches. We feel that evaluating the biological functionality of cells or tissues cultivated within the devices by, for example, immunostaining or genetic sequencing, therefore exceeds the scope of this initial study.

Round 2
Reviewer 2 Report
Comments and Suggestions for Authors
This manuscript was revised well and can be published as it is.